# Conforming Capacitive Load Cells for Conical Pick Cutters

**DOI:** 10.3390/s24134238

**Published:** 2024-06-29

**Authors:** Austin F. Oltmanns, Andrew J. Petruska

**Affiliations:** Mechanical Engineering, Colorado School of Mines, 1500 Illinois St., Golden, CO 80401, USA; apetruska@mines.edu

**Keywords:** capacitive load cell, underground coal mining, conical picks, air-gap capacitive sensor, force sensor, conforming sensor

## Abstract

In underground coal mining, machine operators put themselves at risk when getting close to the machine or cutting face to observe the process. To improve the safety and efficiency of machine operators, a cutting force sensor is proposed. A linear cutting machine is used to cut two separate coal samples cast in concrete with conical pick cutters to simulate mining with a continuous miner. Linear and neural network regression models are fit using 100 random 70:30 test/train splits. The normal force exceeds 60 kN during the rock-cutting tests, and it is averaged using a low pass filter with a 10 Hertz cutoff frequency. The sensor uses measurements of the resonant frequency of capacitive cells in a steel case to determine cutting forces. When used in the rock-cutting experiments, the sensor conforms to the tooling and the stiffness and sensitivity are increased compared to the initial configuration. The sensor is able to track the normal force on the conical picks with a mean absolute error less than 6 kN and an $R^{2}$ score greater than 0.60 using linear regression. A small neural network with a second-order polynomial expansion is able to improve this to a mean absolute error of less than 4 kN and an $R^{2}$ score of around 0.80. Filtering measurements before regression fitting is explored. This type of sensor could allow operators to assess tool wear and material type using objective force measurements while maintaining a greater distance from the cutting interface.

## 1. Introduction

The health and safety of underground coal miners is affected by nearly every aspect of the operation [1]. The effects of these hazards are mitigated by proper ventilation and extraction planning [2]. Machine operators need feedback from the cutting interface to achieve the planned extraction [3], but the cutting interface is the source of harmful dust, gasses, and tunnel conditions. The proposed sensor can collect force data from individual conical picks at the cutting interface, providing operators more feedback while allowing them to maintain a greater distance.

In the U.S.A., employment in the coal mining sector has generally decreased since the 1970s boom, but it experienced a brief renaissance during 2000–2010 [4]. Employment in coal mining in the U.S.A. roughly halved over the period between 2010 and 2020 [5]. The allocation of resources that reduce accidents and increase mine output is the optimal trajectory for advancing technology in underground coal mining [6]. If worker supply continues to decline and regulations increase, boosted worker productivity will be needed to meet future demand for coal.

Coal is generally a soft rock and can be broken apart by the average set of human hands in small quantities. The rock in which coal is often deposited is much harder and is cut by a continuous mining machine with a drum holding an array of conical picks. The specific energy of the cutting process has a strong correlation with material type and tool wear [7]. For this work, a sensor design that measures the forces present on the picks is chosen. These force measurements provide direct and objective feedback from the cutting process.

This work is follow-up research for a previous sensor design, which used a much thinner membrane, 25 μm thick, with nonlinear deformation dynamics, between two electrodes [8]. Here, a design with a much thicker dielectric, with nominal thickness of 240 μm, and a single conductive plate is tested. The force sensor fits under the sleeve of a conical pick and can provide real-time force feedback from the location of the tool. Operators must make decisions regarding tool changes and cutting strategies. This device would assist operators by providing measurements of the rock-cutting forces present at each instrumented pick, allowing them to make decisions from more remote locations.

A proposed network topology for the efficient use of the sensor in an underground mine is shown in Figure 1. This figure shows multiple smart picks connected via CAN bus, which has been recognized as meeting the requirements for other underground coal mining applications [9,10]. The data from the smart picks are aggregated and compressed by an edge processor on the drum, referred to as the drum processor. From the drum processor, data are both sent wirelessly to the operator in real time and to a central storage server, also known as the cloud. The information could be displayed to the operator using a backlit display on the remote controls for the machine. The data in the cloud can be analyzed to identify trends in production efficiency and safety.

Instrumenting each pick, or a representative subset, allows the cutter-head state to be captured, which is necessary for operators to make informed decisions regarding tool change scheduling and machine control. This setup also allows the human operator to make informed decisions using the collected feedback while staying in a safer position. This could aid in identifying if the tool force is much greater than expected, reducing incidences of operators approaching the machine to check tool wear.

This work focuses on the design of the force sensor at the cutting interface. When it comes to designing a force sensor for underground mining applications, several base designs and materials were considered. In general, rock-cutting forces depend on the tool geometry and the material properties, and they can range from dozens to hundreds of kN at each pick [11,12,13,14]. The operation of mining equipment and the complicated network of jagged tunnels cause the environment to be noisy electromagnetically and acoustically [15,16,17,18,19]. The mining environment is also very harsh, and cutting tools can be consumed quickly, typically being consumed after clearing several cubic meters [20,21,22]. These conditions require any sensor to be robust and also low cost.

It has long been known in rock cutting that the specific energy of a material being removed is related to its strength when breaking down [7]. Also, when a conical pick wears down to a rounded shape, the specific energy of the cutting process increases, resulting in less cutting efficiency and higher forces [11]. In an underground coal mine, operators directly use many cues from the environment to determine tool wear levels and material types [3]. Objective feedback in the form of force measurements stands to improve operator efficiency.

Other methods, like vibro-acoustic signal processing, are gaining popularity for tool wear and material classification in domains like metal milling [23] and oil drilling [24]. Acoustic techniques can be susceptible to interference from nearby sources, but a force sensor on the pick will have a direct link to the cutting process. Many different technologies exist for detecting roof fall conditions [25,26,27]. The proposed technology would help mitigate unintentional roof damage by directly measuring cutting tool force and facilitating the rapid response of either operators or machine safety features.

To measure forces in a tough environment with a low-cost sensor, a capacitive sensor enclosed in a steel case with a polyimide dielectric is designed. The following simple parallel plate approximation for the capacitive sensor is used:(1)$$C=\frac{\epsilon A}{d},$$
where *C* is the resulting capacitance, ϵ is the absolute permittivity of the dielectric, *A* is the area of overlap between the plates, and *d* is the distance between them [28]. Fringing effects are small compared to the nominal capacitance at the resting displacement [29]. The sensor is designed so force compresses the sensor, altering the *d* value in (Equation 1) [30,31]. Other sensor designs involve co-planer electrodes [32,33,34], different dielectrics [35], or motion parallel to the plates [36,37,38]. This sensor is designed to be compressed normal to the plates for a robust design that can withstand the forces of rock cutting.

The sensor measurements would be collected and forwarded to operators wirelessly so they can make decisions with more information. Such a device would also be able to log measurements for later analysis and process optimization. The design and validation of a single smart bit sensor is the focus of this article. The rest of this article is organized as follows: The next section is the Section 2, which discusses the experimental setup, the regression techniques used for the empirical models, and the analytical modeling for the sensors. Then, the Section 3 shows data from the calibration and test procedures, sharing empirical values for sensitivity and comparisons of force predictions using the sensor measurements with measurements from the test equipment. Finally, the Section 4 mentions both benefits and drawbacks to the implementation, while the Section 5 shares potential impacts of this technology.

## 2. Materials and Methods

The sensor requirements for force-sensing range, bandwidth, accuracy, and resolution are chosen first. A force-sensing range of 0 to 200 kN is targeted. For bandwidth, the tracking of the average, rather than the instantaneous, cutting force is of interest because tool wear and material type are known to change average cutting forces. Tracking high-frequency changes in cutting force can be useful for the classification of material type and tool wear, as shown in the previous work by the authors [8]. Using lower frequencies for regression de-emphasizes the differences in dynamics between the capacitive sensor and the test equipment sensors caused by their distance and individual designs by focusing on the average forces.

The regression target is chosen to be the drag force after it is filtered with a low pass filter with a 10 Hz cutoff frequency. Making the regression target an average lowers the variance that must be estimated. This relaxes the regression problem, but still gives a sensor with known accuracy and response time. The requirements for accuracy and resolution are not very stringent due to the large differences in forces that can be expected between distinct materials and wear conditions. In terms of mean absolute error, 15 kN or less would enable the basic classification of material and wear conditions.

Finished samples of the sensor design are shown in Figure 2. Two types of experiments are conducted with the sensor, load frame testing and rock cut testing. Both tests use the FDC2114 device from Texas Instruments, headquartered in Dallas, Texas USA, to record the capacitive data from each of the 4 channels at roughly 400 Hz. The load frame test measures the sensor response to gentle load profiles in a controlled environment, and the rock cut test demonstrates use of the sensor for the application.

For the load frame testing, load profiles which ramped up to 200 kN then back down at rates of 2, 4, 6, 8, and 10 kN/s were applied. The relationship between the sensor measurements and the force and strain measurements from the load frame were examined. With this test, the sensor linearity could be measured and the noise could be characterized in a controlled environment. The sensitivity to normal force was measured with a linear regression. One of the samples was tested twice to characterize repeatability. The setup is shown in Figure 3.

For the rock cut testing, the linear cutting machine at the Colorado School of Mines Earth Mechanics Institute, shown in Figure 4, was used. Two samples of coal embedded in concrete were cut using conical picks instrumented with one of the sensors. The wear and depth of the cut were varied for the tests. The conditions for the test and the number of samples for each condition are shown in Table 1. The cutting speed was 25.4 cm/s (10 in./s). In this test, the sensor changed configuration, and empirical models were fit to transform the sensor measurements into normal force on the tool.

For this study, different mathematical models were used to describe the two tests performed with the sensor. For the load frame tests, the sensor sensitivity was low but demonstrated the working principle of the sensor in a controlled environment. For the rock-cutting tests, the sensor sensitivity was much greater. This can be explained by plastic deformation closing the air-gap area. The models before and after this event are referred to as the air-gap and closed-area models, respectively. The following analytical models describe how this transition can increase sensitivity.

Due to the large number of design parameters, data-driven methods were leveraged to formulate the relationship between measurements and normal force on the tool. The analytical models show that the system was nonlinear, so second-order polynomial expansion was used to allow the regression models to compensate more accurately. The analytical models for both configurations are described in the following subsection. After that, the next subsection discusses the regression techniques used to estimate the cutting drag force from the sensor measurements.

### 2.1. Analytical Modeling

To model the relationship between input force and measured resonant frequency, the sensing element and the circuit inductance and capacitance is considered. Slight parasitic capacitance and inductance was present when comparing the analytical model to the experimental results. The FDC2114 Capacitance to Digital Converter was used to measure the resonant frequency of all four sensor channels. This setup had a resolution of 2.44 kHz per bit and used a reference frequency of 40 MHz and an internal gain setting of 2. With this configuration, the converter could measure resonant frequencies up to 4 MHz.

The resonant frequency of the sensing circuit was that of a parallel inductor and capacitor [39]:(2)$$f=\frac{1}{2\pi \sqrt{L_{b}\left[C_{b}+C_{s}\right]}},$$
where $L_{b}$ is the lumped inductance from the board and parasitics, $C_{b}$ is the lumped capacitance from the board and parasitics, and $C_{s}$ is the capacitance for the sensor element, which is electrically parallel to $C_{b}$. This model was used to explore the trade-offs in performance and changes in sensitivity between the two models. To describe the different design aspects of the sensor performance, the chain of derivatives from resonant frequency to input force,
(3)$$\frac{\partial f}{\partial F_{in}}=\frac{\partial f}{\partial C_{s}}\;\frac{\partial C_{s}}{\partial d_{r}}\;\frac{\partial d_{r}}{\partial F_{in}},$$
is helpful. The significance of each term on the right hand side is explained below.

It was found during calibration and testing that it was necessary to model both the air-gap sensor and a sensor with a closed area. The model with the air gap explained the characterization experiment with good accuracy, while the model with the closed area explained the rock-cutting experiment with good accuracy. The closing of the air gap suggests that rock cutting produces large forces on the cutting tools. The character of the sensor deformation can be seen in Figure 2. Model diagrams for both modes of sensor operation are shown in Figure 5. The first term on the right hand side of Equation (Equation 3), the sensitivity of frequency to sensor capacitance, is the same for both of the sensor configurations. The other two terms are unique to each model.

The sensitivity of the sensor frequency with respect to sensor capacitance is as follows:(4)$$\frac{\partial f}{\partial C_{s}}=\frac{-L_{b}}{4\pi {\left(L_{b}\left(C_{b}+C_{s}\right)\right)}^{\frac{3}{2}}}.$$
If the other sensitivities, $C_{b}$, and $L_{b}$ are held constant, larger $C_{s}$ values will give less sensitive sensors. The second derivative has a similar shape, indicating that while less sensitive, larger $C_{s}$ values will have more linear responses. To understand how the sensor responds to strain, the next term, the sensitivity of capacitance to displacement, is needed.

For the air-gap sensor, $r_{3}$ is the region where thickness varies with the input force. When considering the closed-area sensor, both $r_{1}$ and $r_{2}$ will deform, with most of the deformation happening in $r_{1}$ since it is less stiff in comparison to the much thinner $r_{2}$. The air-gap sensor is less stiff than the closed-area sensor, as the polyimide is much stiffer than the thin steel walls of the sensor. Nominal values for the design are shown in Table 2. Analytical models for both configurations are derived below, and empirical values for sensitivity are shown in the Section 3.

#### 2.1.1. Air Gap

For the isolated sensor with an air gap, the model of input force to capacitance is derived as follows: Each capacitive cell of the sensor is modeled as having three regions. The first region is between the electrode and the bottom of the steel case, consisting of polyimide and a thin layer of soldermask, roughly 25 μm. The second and third regions are above the electrode and are the soldermask and air regions respectively. The first region is electrically parallel to the second and third regions, which are in series with each other.

When force is applied, the sensor case compresses, decreasing the thickness of $r_{3}$. The overall capacitance of the sensing element, $C_{s}$, can then be modeled as follows:(5)$$C_{s}=C_{r_{1}}+{\left[\frac{1}{C_{r_{2}}}+\frac{1}{C_{r_{3}}}\right]}^{-1},$$
where each $C_{r_{n}}$ is the individual capacitance of the region and $C_{r_{3}}$ is the variable air-gap region. To calculate the sensitivity capacitance with respect to displacement, $C_{r_{3}}$ is expanded with Equation (Equation 1) and partially differentiated with respect to $C_{s}$, yielding
(6)$$\frac{\partial C_{s}}{\partial d_{r_{3}}}=\frac{-\epsilon_{r_{3}}AC_{r_{2}}^{2}}{{\left[\epsilon_{r_{3}}A+C_{r_{2}}d_{r_{3}}\right]}^{2}}.$$
Then, to find the upper bound of the sensitivity, the limit as the gap size approaches zero is found, simplified, and rearranged:(7)$$\begin{array}{c} \lim_{d_{r_{3}}\to 0}\frac{-\epsilon_{r_{3}}AC_{r_{2}}^{2}}{{\left[\epsilon_{r_{3}}A+C_{r_{2}}d_{r_{3}}\right]}^{2}}=\frac{-\epsilon_{r_{3}}AC_{r_{2}}^{2}}{{\left[-\epsilon_{r_{3}}A\right]}^{2}}=\frac{-C_{r_{2}}^{2}}{\epsilon_{r_{3}}A}=\frac{-\epsilon_{r_{2}}}{\epsilon_{r_{3}}}\frac{C_{r_{2}}}{d_{r_{2}}}. \end{array}$$
The sensitivity is upper bounded in magnitude by the product of the ratio of dielectric permittivities and the ratio of capacitance to thickness for $r_{2}$.

The thickness of $r_{3}$, denoted as $d_{r_{3}}$, as a function of the input force, $F_{in}$, can be modeled as
(8)$$d_{r_{3}}\left(F_{in}\right)=d_{g}-F_{in}/K_{w},$$
where $d_{g}$ is the nominal thickness of the air-gap region and $K_{w}$ is the stiffness of the steel case walls. Sensor stiffness is a direct trade-off for sensitivity. A stiffer sensor will be less sensitive, but will likely have more linear performance by limiting the range of displacement and capacitance. A thinner sensor will be more sensitive, as the same distance of displacement will cause much larger changes in capacitance. A very thin and stiff sensor would then have good sensitivity and linearity can be achieved by tuning the overall stiffness.

#### 2.1.2. Closed Area

Considering a closed air-gap area means eliminating the third region from the electrical model and adding the stiffness of the flexible dielectric to the sensor’s physical model. In this mode, both regions should experience some deformation, but the thicker region is likely to be more sensitive since it should be less stiff. The sensor capacitance is given as the sum of the two regions:(9)$$C_{s}=C_{r_{1}}+C_{r_{2}}.$$
The sensitivity to displacement, after substituting in for $C_{r_{1}}$ and $C_{r_{2}}$ is then
(10)$$\left[\begin{array}{c} \frac{\partial C_{s}}{\partial d_{r_{1}}}\\ \frac{\partial C_{s}}{\partial d_{r_{2}}} \end{array}\right]=\left[\begin{array}{c} \frac{-\epsilon_{r_{1}}A}{d_{r_{1}}^{2}}\\ \frac{-\epsilon_{r_{2}}A}{d_{r_{2}}^{2}} \end{array}\right],$$
and the deformations for the two regions are modeled as follows:(11)$$\begin{array}{cc} d_{r_{1}}\left(F_{in}\right) & =d_{{r_{1}}_{n}}-\frac{F_{in}}{k_{1}} \end{array}$$(12)$$\begin{array}{cc} k_{1} & =\left[K_{w}+{\left[1/K_{r_{1}}+1/K_{r_{2}}\right]}^{-1}\right]\left[K_{r_{2}}/K_{r_{1}}+1\right] \end{array}$$
(13)$$\begin{array}{cc} d_{r_{2}}\left(F_{in}\right) & =d_{{r_{2}}_{n}}-\frac{F_{in}}{k_{2}} \end{array}$$
(14)$$\begin{array}{cc} k_{2} & =\left[K_{w}+{\left[1/K_{r_{2}}+1/K_{r_{1}}\right]}^{-1}\right]\left[K_{r_{1}}/K_{r_{2}}+1\right] \end{array}$$

With the given Young’s modulus and height for $r_{1}$ and $r_{2}$, the flexible printed circuit can be expected to be have a total stiffness of around 80 giganewtons/meter. The soldermask region stiffness, $K_{r_{2}}$, has a calculated value of roughly 275 GN/m, and the polyimide and soldermask region, $K_{r_{1}}$, has a calculated stiffness of roughly 116 GN/m. Increases in stiffness for this sensor configuration result in a more linear sensor than one that is less stiff. Like the air-gap sensor, the closed-area sensor sensitivity still varies with the strain of the sensor.

The sensitivities of the two models are compared by rewriting Equation (Equation 10) using the capacitances of the regions in the air-gap sensor to give
(15)$$\left[\begin{array}{c} \frac{\partial C_{s}}{\partial d_{r_{1}}}\\ \frac{\partial C_{s}}{\partial d_{r_{2}}} \end{array}\right]=\left[\begin{array}{c} \frac{-C_{r_{1}}}{d_{r_{1}}}\\ \frac{-C_{r_{2}}}{d_{r_{2}}} \end{array}\right],$$
Depending on the shift in nominal values for $d_{r_{1}}$ and $d_{r_{2}}$, the closed-area configuration can have much greater sensitivity than the air-gap model. The air-gap configuration sensitivity is upper bound to a constant times the ratio $C_{r_{2}}/d_{r_{2}}$, with that constant being the ratio of dielectric permittivities between the soldermask and air. In the closed-area configuration, if $d_{r_{2}}$ is reduced by half, due to plastic deformation in this case, the capacitance will double and result in four times more sensitivity in that term alone. The ratio of permittivities is roughly between two and five, so the additional sensitivity from adding $r_{1}$ and compression of the sensor is likely to make the closed-area configuration more sensitive compared to the air-gap configuration.

### 2.2. Regression Techniques

The methods used to transform the sensor measurements to drag force estimates in the rock-cutting experiment are linear regression [40] and neural networks with rectified linear units for the activation function [41,42]. For both methods, two sets of coefficients are used: the 4 channels of the sensor and the 2nd-order polynomial expansion of the four channels. For each of the four regression methods, it was found that the higher frequency inputs were not correlated to the regression target. So, the effect on performance of low pass filters with different cutoff frequencies used on the input were also tested. A summary of the chosen regression methods and their size is shown in Table 3.

Each of these models were compared using 100 random 70:30 test and train splits, a technique known as Monte Carlo cross-validation [43]. Problems can arise when using this method to validate the classification of features that are rare in a data set [44,45]. A large number of splits was used, with only 30% of data used for training in each split, to try to accurately capture the performance distributions. The input dimension was small, either 4 or 13 values, compared to the number of samples, 46,201 data points. A good regression fit would demonstrate the linear performance of the sensor for average force estimation and would show that the performance in this test would generalize to an expanded dataset.

The regression methods are compared on the basis of mean absolute error and $R^{2}$ score [46,47,48]. The pursuit of a best metric is often debated, and the selection must be applied appropriately [49]. The root mean squared error and the $R^{2}$ scores will provide the same overall ordering for the methods. The mean absolute error will give less of a penalty to outliers than the $R^{2}$ score [50]. Since instantaneous readings of the four sensor channels were used, the input dimension was small and the $R^{2}$ score did not need adjustment [48]. The $R^{2}$ score is a good proxy for force tracking, while the mean absolute error is a good metric for accuracy. Using a collection of metrics promotes a better qualitative analysis of the regression models’ performances.

Each method is framed as trying to solve for the instantaneous normal force on the tool, *y*, from a vector of sensor measurements *x*. This measurement vector for the instantaneous linear case is written as $x={\left[a,b,c,d\right]}^{⊤}$, where a,b,c, and *d* represent the values from the four sensor channels at that moment. The channel values are the measured change in resonant frequency from right before the current cut. For the polynomial case, *x* is expanded with the unique second-order pairings as $x=[a,b,c,d,a^{2},b^{2},c^{2},d^{2},ab,ad,bc,bd,cd]$.

The linear regression method aims to find a set of coefficients $\left\{L_{1},\ldots ,L_{n}\right\}$, where *n* is the size of the input dimension, and a scalar bias, *B*, such that
(16)$$y=\sum_{i=1}^{n}L_{i}x_{i}+B.$$
Given a finite set of input–output pairs, the optimal $L_{i}$ and *B* values that generate the least error according to different metrics can be calculated in closed form [40]. This regression method is robust in the sense that it has few parameters, can operate off a small number of input variables, and will have predictable output for all inputs.

The neural network regression models take a similar approach to the linear regression. The neural network consists of layers of neurons, where each neuron in each layer applies an activation function to a weighted sum of the outputs from the previous layer. The rectified linear activation function [41] is used, which has the following form:(17)$$\sigma \left(z\right)=\left\{\begin{array}{cc} 0, & z\leq 0\\ z, & \mathrm{otherwise} \end{array}\right..$$
This activation function allows nonlinear relationships to be modeled by the neural network.

The number of neurons in a layer is the width, and the number of layers of neurons is the depth. Consider a network with depth *D*; we denote the width of each layer as $W_{d}$ for d∈{1…D}. For a single neuron at layer *d* and position $t\in 1\ldots W_{d}$, with the previous layer having width $W_{d-1}$, the output is
(18)$$s_{d,t}=\sigma \left(\sum_{i=1}^{W_{d-1}}L_{i,t}s_{d-1,i}+B_{d,t}\right).$$
where σ(·) is the activation function, $L_{i,t}$ represents the learned coefficients, $s_{d-1,i}$ are the output from either the neurons in the previous layer or the input data vector entries for the first layer, and $B_{d,t}$ is the learned bias for the neuron. The output of the neural network is a weighted sum of the outputs of the neurons in the last layer with no activation function:(19)$$y=\sum_{i=1}^{W_{D}}L_{i,t}s_{D,i}+B_{D}$$

This means that each neuron has roughly the same number of parameters as the entire linear regression problem. A large enough network such as this can memorize a finite data set [51]. A smaller network is more general, while a larger network will begin to highlight any biases in the collected data. Unlike the linear regression problem, solving for the neural network coefficients is carried out via an iterative process. To limit the number of parameters from being too large, a network size of 3 hidden layers is chosen, with the width being the same size as the input.

The experiments were implemented using scikit-learn, also known as sklearn [52], and ran on a laptop computer. The regression experiments took a couple of hours to run. The code is available at https://github.com/Fworg64/air_gap_coal_sensor_model (Published by the authors of this paper on 28 April 2024, see tag v1.0). The analytical models revealed that the system is nonlinear and has polynomial terms. This analysis guides the selection of empirical models and preprocessing methods.

## 3. Results

The sensitivity of each configuration was compared empirically based on the results. The slope magnitude for the linear regression of the air-gap characterization test was 10.164 kN/kHz, and so for each kilonewton of force, the resonant frequency dropped by about 98.4 Hz. The sensitivity of the closed-area sensor was derived using the average of the coefficients from the linear model trained on the rock-cutting data. The average of the crushed gap values was 0.1522 kN/kHz, which means that each kilonewton of force reduced the resonant frequency by roughly 6.57 kHz. The values for both models are shown in Table 4.

The closed-area configuration was over 65 times more sensitive than the air-gap sensor. The air-gap sensor had about eight levels over the 200 kN input range. With the closed-area sensitivity and the capacitance to digital converter resolution of 2.44 kHz, the crushed gap had roughly 80 levels over the same range, giving it 10 times the resolution. Using Equation (Equation 10), it is inferred that $d_{r_{1}}$ and $d_{r_{2}}$ were crushed to a fraction of their original thicknesses.

### 3.1. Air-Gap Load Frame Characterization

The results from the load frame characterization of the sensor show that the response was noisy but linear over the tested range. In the repeated test, the sensor had almost identical performance but without the initial plastic deformation in the 2 kN/s test. The strain of the sensor for each test, shown in Figure 6, was nearly identical after the initial plastic deformation. The plot shows the significant deformation of one of the test sensors during its first loading cycle. After this first loading cycle, the device strain for each test sensor was similar. The peak force of each test was 200 kN, and the sensor consistently deformed with a strain of 0.14 at this peak. Considering a sensor height of 1.83 mm, the stiffness, $K_{w}$, was roughly 780 MN/m.

The load frame force measurements and the sensor measurements from the test for the air-gap sensor are shown in Figure 7. The test suite was repeated for the prototype that was not used in the rock test, and the measurements were consistent between the tests. There was some plastic deformation of the steel case during the initial loading of the sensor in the 2 kN/s case. After this, the sensor had a mostly symmetric response to loading and unloading. The two sensors had similar sensitivity but different offsets after the plastic deformation phase.

An XY plot of the traces is shown in Figure 8, which highlights the linear range and repeatability of this configuration. The 2 kN/s tests are not representative due to their large, one-time swing in values caused by the initial plastic deformation. They were excluded from this graph and the sensitivity calculation. There was some measurement creep, but ramping the input force to 200 kN would increase the digital sensor measurement roughly 5 to 8 levels. The bias for the resonant frequency was reset at the beginning of each test just before force was applied. Compared to the first sensors 4 kN/s test and the second sensors first 10 kN/s test, most of the other tests had similar measurements.

### 3.2. Rock Cutting and Model Fitting

The initial sensor design was for the air-gap configuration. The sensitivity as measured by the load frame test was deemed sufficient for the binary classification of hard or soft rock. The fact that the sensor compressed into a more sensitive version under the rock-cutting forces demonstrates that a sensor must be robust to the large forces in this application. The toughness of the steel and polyimide, and their increased stiffness after compression, resulted in a usable sensor.

For the rock-cutting experiments, the linear and neural network models previously discussed in Section 2.2 were fit to the data. The large number of parameters in the physical model made these regression techniques a good fit for estimating the cutting force. Prediction results for a few samples with new picks are shown in Figure 9. More prediction results using the worn tool are shown in Figure 10.

The strain gauge measurements had large variance due to the rock chipping at high frequency. The regression target is highlighted in magenta. The different regression methods tracked the force as the tool cut through the sample. The middle of the sample was coal, and this generally takes less force to cut than the surrounding concrete. The sensor could be used to identify changes in material based on differences in cutting force. The use of the worn tool caused greater peak force values in the experiment. The sensor was able to track the force with different wear conditions using each of the regression techniques. The sensor could be used to detect changes in tool wear based on increases in cutting force when cutting the same material.

Using a low pass filter on the sensor measurements improved performance. The mean absolute error for each classification method using the different low pass filters is shown in Figure 11. The $R^{2}$ scores are shown in Figure 12. Using a cutoff frequency of 10 Hz or 5 Hz gave the best results across methods. The neural network regression method with the second-order polynomial expansion performed the best overall.

By filtering the measurements before fitting the regression, the tracking error was reduced. The performance of the second-order polynomial feed-forward neural network regression broke away from the rest due to its greater capacity for nonlinear modeling. Using a cutoff frequency the same or slightly lower than the regression target gave the best results. The filtering was needed to make the best-performing method reliable, as the control case gave some regression methods which did not function well for all splits. The regressions without neural networks gave very consistent performances.

The analytical models show that the relationship between resonant frequency and input force is nonlinear, using hyperbolic and inverse square root terms. Use of the second-order polynomial expansion improved results for both the linear regression and the neural network regression. Use of more sophisticated methods like the neural network regression method significantly improved performance when used with the second-order polynomial expansion. The neural network without the expanded input was less reliable, with some fraction of the trained classifiers always giving a bad performance.

To show that the signals measured by the sensor could be used for tool wear and rock type classification, a classic two-sided Welch–Satterthwaite *t*-test [53] was performed on each frequency bin after transformation with a short-time Fourier transform. This method was shown in our previous work, [54], which used short-time Fourier spectra coefficients of acoustic data for tool wear classification with conical picks, to be a good predictor for the ability to classify signal samples into their relevant category. For material classification, the distribution of Fourier spectra coefficients is shown in Figure 13. For tool wear classification, the distribution is shown in Figure 14.

In both plots, the square root of the coefficients is shown on a log scale for convenient plotting. The frequency bins with significant, p<0.01, differences in statistical distribution are highlighted. These differences confirm that different tool wear and material conditions generate force signals according to different distributions. This fact implies that the signals could be classified using methods such as the support-vector machine, as shown in [8,54]. When comparing the two plots, it can be seen that more of the coefficients from the different material categories have more significant differences than the coefficients from the different wear categories. This implies that the classification of coal vs. concrete, or other hard rock, may be easier to perform accurately than the classification of new tools vs. worn tools when using data collected from this sensor.

The capacitance to digital converter measured the resonant frequency of the sensing cell, which varied with the pressure on the cell. Considering force normal to the sensor, the area of contact was nearly constant, and the sensor’s resonant frequency varied with force on the tool. This change in the sensor resonant frequency was used to deduce the force on the tool. This sensor measured the force on the tool, as this correlated strongly with tool wear and material type without needing to deduce the precise area of the tool in contact with the material. Sensor sensitivity is key to accurately determining these forces, and in our previous work [8], we developed a sensor for rock cutting which had a sensitivity of roughly 1.0 ticks/kN, while the one developed in this work increased that sensitivity to roughly 2.7 ticks/kN, where a tick is a unit increase in the digital value read by the capacitance to digital converter.

## 4. Discussion

The average normal force was tracked for this study, as this quantity is known to be correlated with both tool wear and material type. The signal was averaged using a low pass filter with a cutoff frequency of 10 Hz, to allow the higher frequency rock chipping forces to be averaged together while still allowing a quick response for material and wear changes. When restricting the regression target bandwidth, the overall variance of the signal was reduced. The higher-frequency components were not needed to track the average.

The sensor sensitivity could be improved by using thinner dielectric regions, but care must be taken to stay within the linear deformation range for the polyimide material. Polyimide has linear characteristics for small deformations but is known to experience hysteresis and temperature dependence [55,56,57]. The deformation characteristics of thin-film polyimide sheets after compression to a fraction of their original height should be investigated and compared to uncompressed sheets of the same height.

The air-gap configuration was useful for gentle load profiles and could have applications outside of underground mining. In the rock-cutting experiment, the polyimide likely deformed to be much thinner than it was initially. For the closed-area configuration, the sensitivity of capacitance to distance increased to infinity as the layers became thinner. The closed-area configuration was more stiff than the air-gap configuration, which helped the sensor keep its linear performance.

This process of conforming to the target tooling is desirable, as it both increases sensor sensitivity and removes ingress opportunities for environmental contaminants to the sensing area. When the sensor conforms to the cutting tool during rock cutting, the air-gap is removed, leaving a nearly sealed layer of polyimide. In an application, the case can be sealed further, preventing the ingress of contaminants. Further testing is required to determine performance at different operating temperatures, but after characterization, one could compensate for the changes by using a thermocouple to measure local tool temperature while maintaining low power operation for the device.

Changes in environmental humidity are likely in an underground mine, which will affect the nominal value of the sensor readout, and higher humidity will increase the dielectric constant of the material and increase the sensor sensitivity. However, the part of the frequency response caused by rock type or tool wear will dominate this effect. For example, the coal will remain much softer than the surrounding rock, even in high-humidity conditions. Using a frequency-based classification method would be a robust way to make these types of classification while tolerating many environmental parameters which can affect the bias of the sensor.

## 5. Conclusions

The sensor performed well when estimating the cutting force, even when using the simple linear regression model. The ability to track cutting force with a sensor can improve operator performance and safety by giving them objective feedback while they maintain a safe distance. Worker efficiency can be increased via the addition of autonomous process control enabled by sensors on continuous miner cutter-head picks. We have validated our sensor under laboratory conditions and believe this technology is ready for the next stage of integration with the target application.

Capacitive sensors are a promising technology for applications which require low power and low cost. This work has shown a design of and validation for a capacitive sensor which conforms to the conical pick cutters used in underground mines and has produced a sensor with demonstrated force measurement capabilities. We also demonstrated that the signals captured by this sensor come from distinct distributions when the tool wear or material types are distinct, which should facilitate the use of this sensor for the classification of these measurements. This design should be easily adaptable to other robotic applications. Our work shows the implementation of a single sensor. Forming a network of these sensors would allow the full cutter-head state to be known without stopping operations or requiring the operator to get close to the cutting interface, improving operator safety and efficiency. 

## Figures and Tables

**Figure 1 sensors-24-04238-f001:**
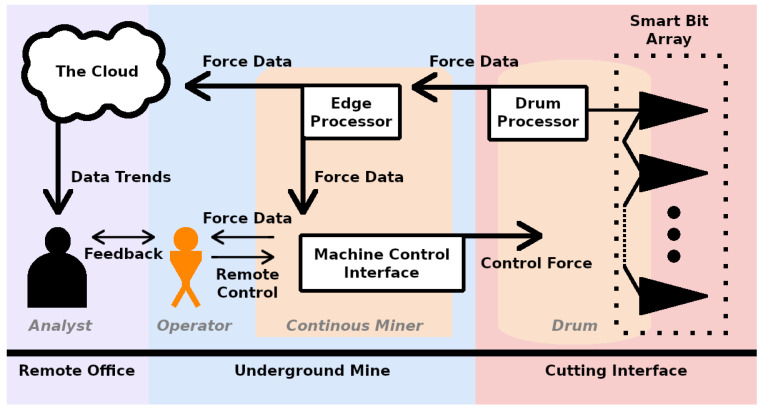
The network proposed for integrating our sensor with the continuous miner. Individual sensors can be linked to an edge processor on the drum which aggregates the force data via CAN bus, a robust interface for sensor networks. From the edge processor on the drum, labeled “Drum Processor”, force data can be sent wirelessly to an edge processor on the machine chassis. There, the data can be parsed into a displayable format for immediate use by the machine operator and also sent up to the surface to be stored in a central server for processing by a mine analyst to gain insight into operator performance. The analyst and the operator can then communicate to increase efficiency and safety.

**Figure 2 sensors-24-04238-f002:**
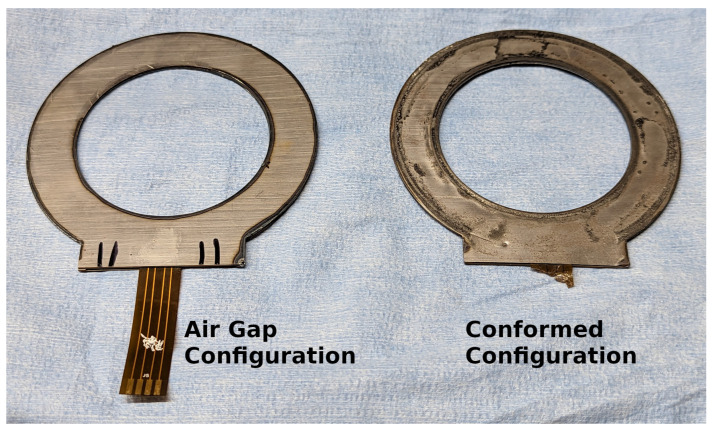
Two sensor prototypes. The one on the left has only been used for characterization in a load cell with controlled parameters. The one on the right has been through the same calibration as well as a controlled rock-cutting experiment. During the rock-cutting experiment, the air gap of the sensor was crushed out, altering the model but still giving a mostly linear sensor. The exposed sensor membrane was inadvertently cut by the edge of the second sample on the last cut. The walls of the sensor retained much of their thickness, but they deformed slightly to match the tooling.

**Figure 3 sensors-24-04238-f003:**
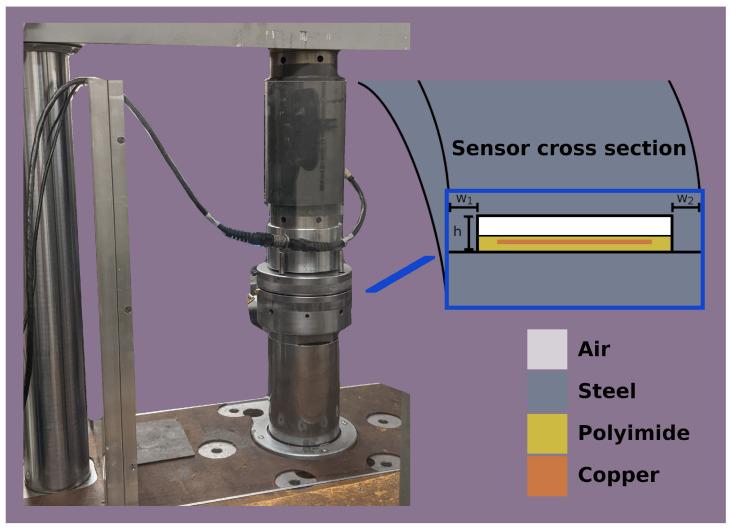
The setup for the load frame characterization of the sensor in the air-gap configuration, shown in the sensor cross-section. The load frame applies a controlled loading profile to the sensor, allowing the response of the sensor to be compared against a controlled input. The load frame measures both force and displacement, while the sensor’s capacitance is measured by the interface circuit. Forces in this test ramp up to 200 kN and back down at a controlled linear rate.

**Figure 4 sensors-24-04238-f004:**
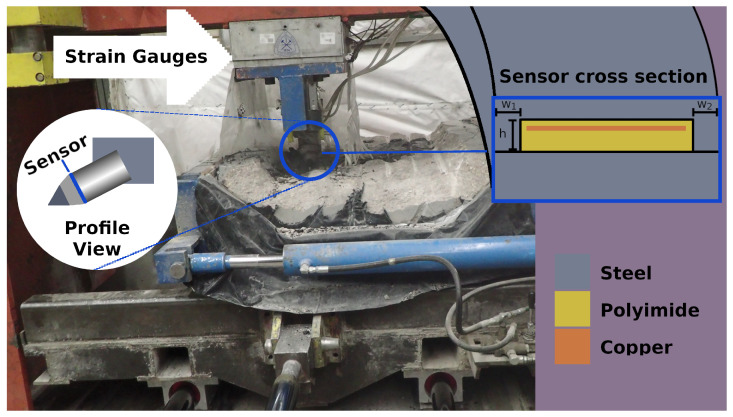
The setup for the rock-cutting experiment. Strain gauges measure the forces on the cutting tool from close proximity while hydraulic actuators drag the rock sample against the cutting tool. The sensor is located between the sleeve and the block of the cutting tool, as shown in the profile view diagram. In this setting, the sensor is in the crushed gap configuration, shown in the sensor cross-section. Forces in this test are less than 100 kN, but the rate of change in force is large and variable.

**Figure 5 sensors-24-04238-f005:**
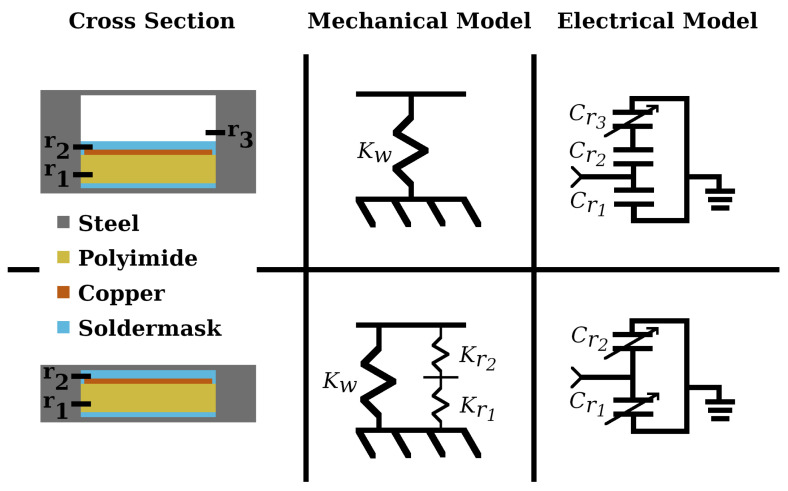
Cross-section models for the two sensor modes. The top model has the air gap intact, resulting in one region which has variable capacitance. The bottom model has a closed air-gap area, resulting in a stiffer sensor with two regions of variable capacitance. The top model is valid for sensors which have not undergone significant plastic deformation, while the bottom model is accurate for sensors after they have formed to the cutting tool. The measured stiffness of the case in the air-gap configuration, $K_{w}$, is around 780 meganewtons/meter. In the closed-area configuration, the polyimide contributes additional stiffness, as it occupies more than twice as much area as the steel walls and is very thin in comparison. The soldermask and polyimide are lumped together for the model of $r_{1}$ since the soldermask is thin in comparison to the polyimide and has similar properties.

**Figure 6 sensors-24-04238-f006:**
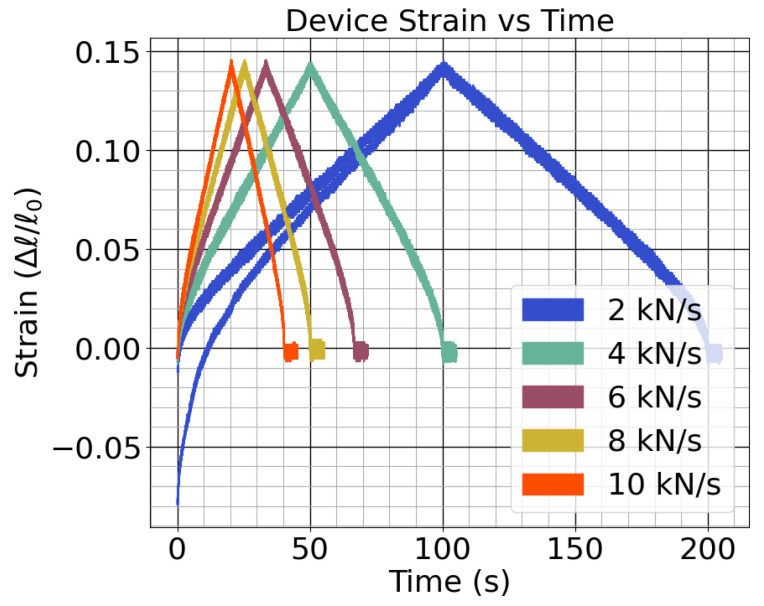
The strain of the physical sensor cases during the tests.

**Figure 7 sensors-24-04238-f007:**
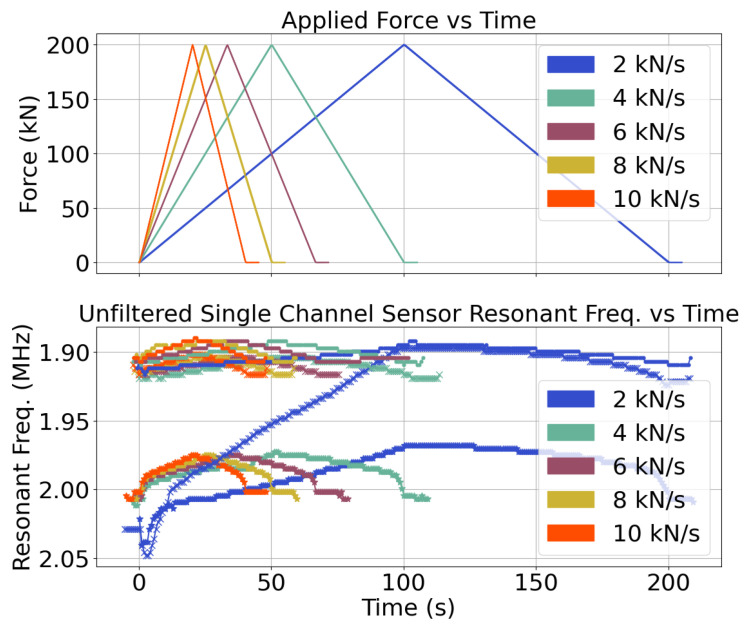
Force profiles and capacitive measurements during loading for the two test sensors.

**Figure 8 sensors-24-04238-f008:**
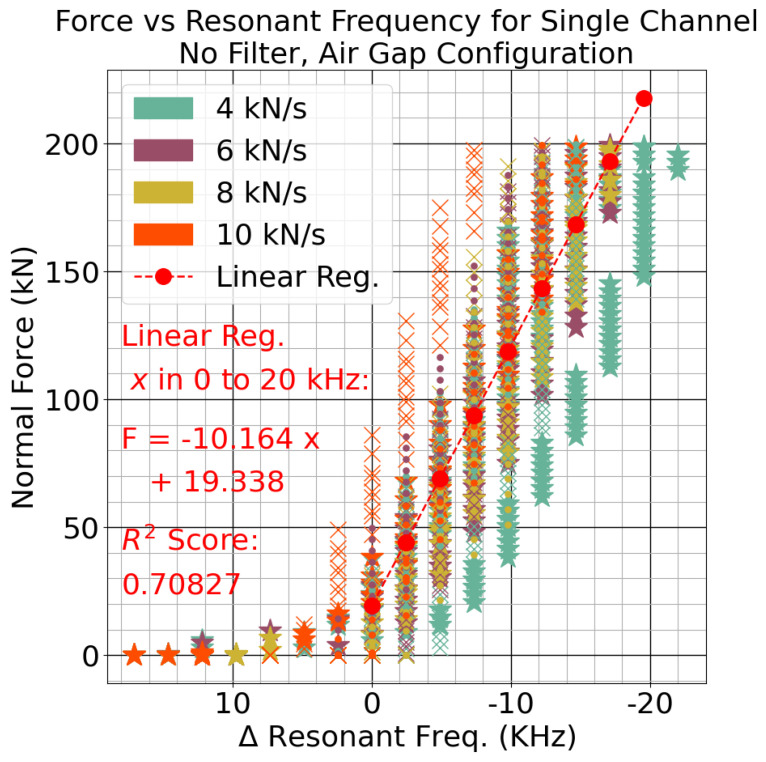
The relationship between measured resonant frequency and applied force for each sensor.

**Figure 9 sensors-24-04238-f009:**
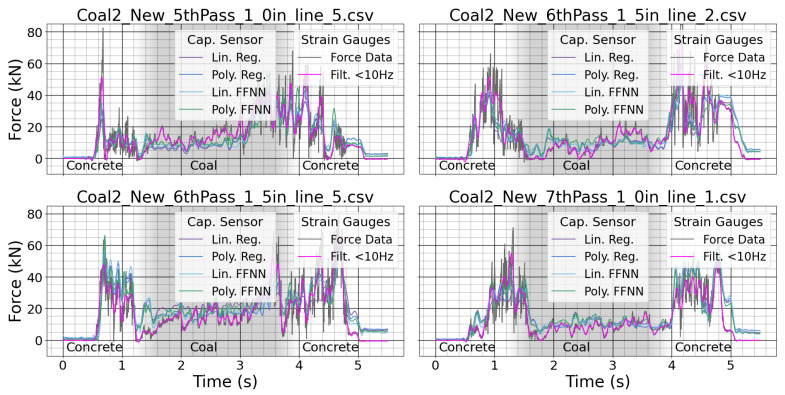
Measurements from both the system strain gauges and the custom sensor for the new tool.

**Figure 10 sensors-24-04238-f010:**
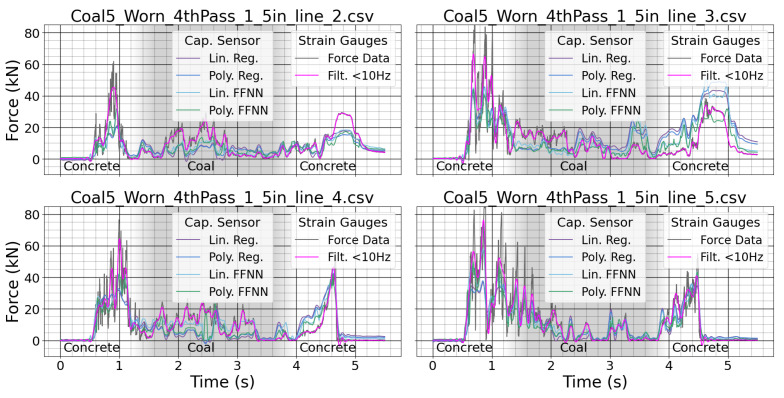
Measurements from both the system strain gauges and the custom sensor for a worn tool.

**Figure 11 sensors-24-04238-f011:**
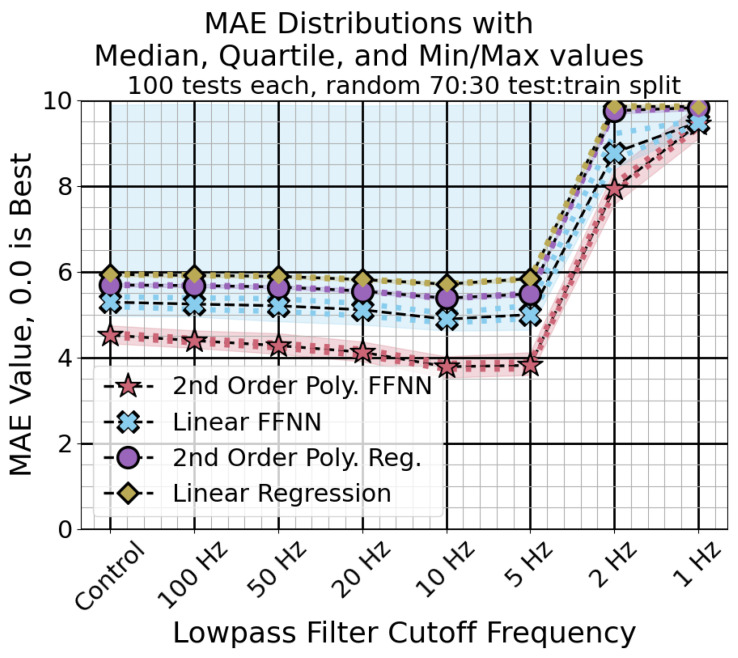
Mean absolute error distributions for different input filter conditions. The markers and dashed lines represent the median, while the dots and shaded area represent the quartile and min/max values, respectively. Lower scores are better.

**Figure 12 sensors-24-04238-f012:**
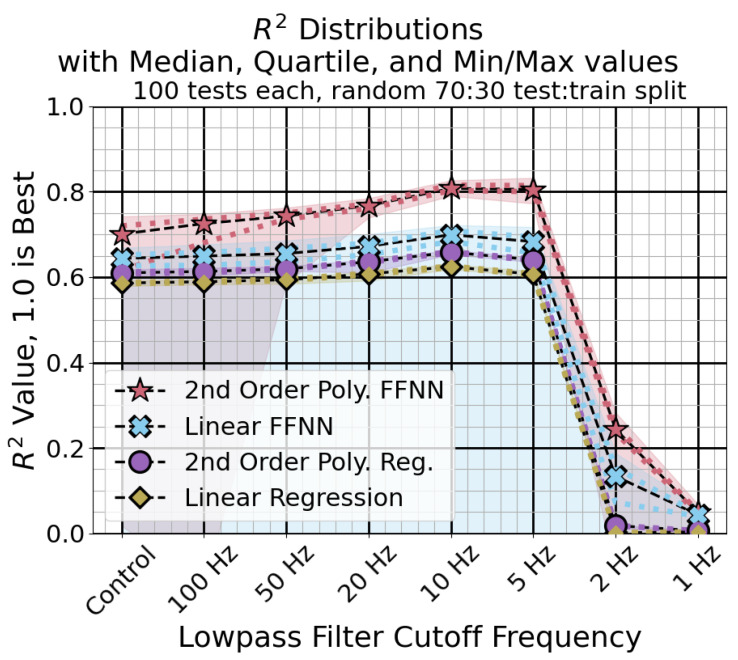
$R^{2}$ score distributions for different input filter conditions. The markers and dashed lines represent the median, while the dots and shaded area represent the quartile and min/max values, respectively. Higher scores are better.

**Figure 13 sensors-24-04238-f013:**
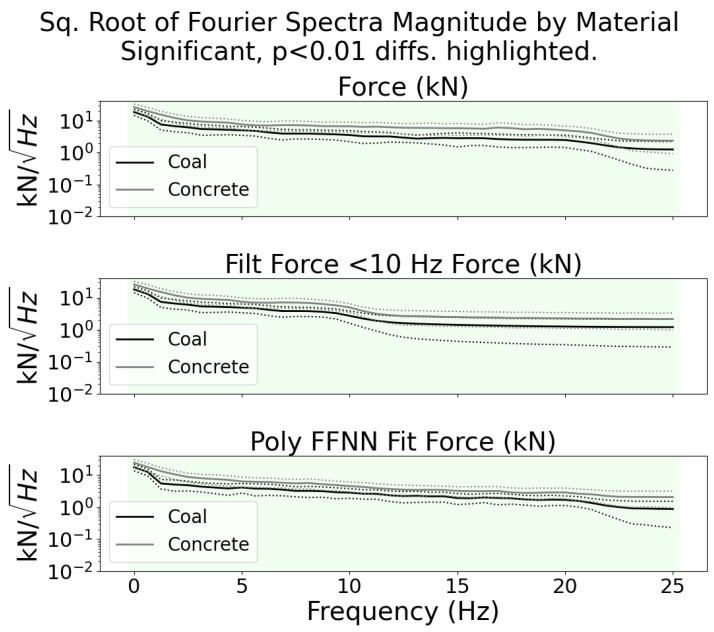
Scaled Fourier spectra magnitude coefficients by material type. The mean value is plotted with a solid line, while the dashed lines represent ± one standard deviation. The distribution of every frequency bin is significantly different between the two material types. This indicates that the classification of the signals could be performed with a good degree of accuracy.

**Figure 14 sensors-24-04238-f014:**
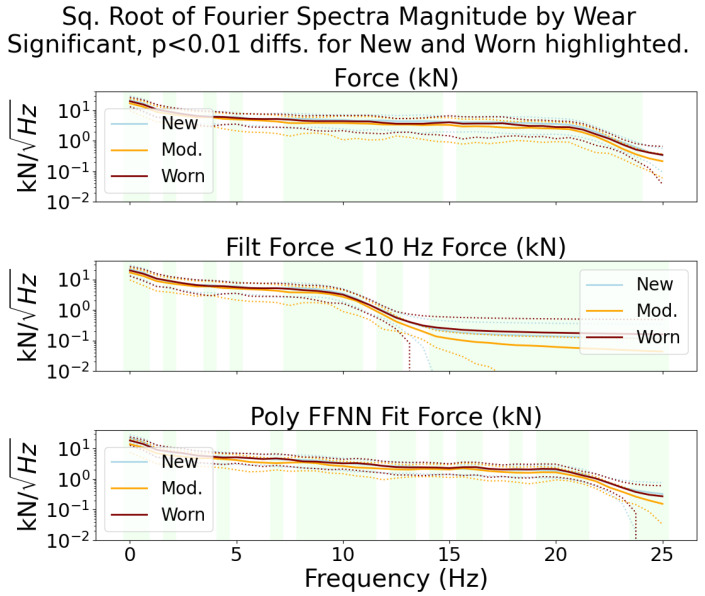
Scaled Fourier spectra magnitude coefficients by tool wear. The mean value is plotted with a solid line, while the dashed lines represent ± one standard deviation. Roughly half of the frequency bins have significant differences between the new and worn categories. This indicates that the classification of tool wear using these signals is possible, but might not be as good as in the material type classification case.

**Table 1 sensors-24-04238-t001:** Rock cut conditions tested, coal samples, and conical picks.

Depth of Cut	Wear Condition (Tip Diameter)	Number of Cuts
1.0 in.	New (3.71 mm)	10
1.5 in.	New (3.71 mm)	2
1.5 in.	Mod (17.9 mm)	4
1.5 in.	Worn (27.5 mm)	5
	Total	21

**Table 2 sensors-24-04238-t002:** Nominal design values for sensor; crushed gap reduces *h*, $d_{r_{1}}$, and $d_{r_{2}}$.

Geometric Parameter	Value	Electrical/Material Property	Value
$d_{r_{1}}$, Height of $r_{1}$	127 μm	$\epsilon_{r_{1}}$, Relative Permittivity of $r_{1}$	3.2 *
$d_{r_{2}}$, Height of $r_{2}$	25 μm	$\epsilon_{r_{2}}$, Relative Permittivity of $r_{2}$	2 to 5 †
$d_{r_{3}}$, Height of $r_{3}$	240 μm	$\epsilon_{r_{3}}$, Relative Permittivity of $r_{3}$	1.0
*h*, Gap Height	457 μm	$K_{w}$, Steel case stiffness	780 MN/m ‡
*A*, Plate Area	3.8 cm^2^	Young’s Modulus Polyimide	7.1 GPa *
$w_{1},w_{2}$, Wall Widths	2.4 mm	Young’s Modulus Soldermask	2.4 GPa **
Inner Diameter	64.0 mm	$L_{b}$, Board Inductance	18 μH
Outer Diameter	96.5 mm	$C_{b}$, Board Capacitance	32 pF

*: Panasonic Felios F775 Polyimide Datasheet. †: Not published, typical value range given here. ‡: Measured with load frame experiment. **: Taiyo America PSR-9000 FXT Series Datasheet.

**Table 3 sensors-24-04238-t003:** Regression methods and number of inputs and trainable parameters.

Regression Technique	# Input Variables	# Parameters
Linear Regression of channel values	4	5
2nd-Order Polynomial Regression	13	14
Neural Network with channel values	4	65
Neural Network with 2nd-Order Polynomial	13	645

**Table 4 sensors-24-04238-t004:** Model values for linear regression coefficients and bias terms.

Sensor Configuration	$L_{1}$ (N/Hz)	$L_{2}$ (N/Hz)	$L_{3}$ (N/Hz)	$L_{4}$ (N/Hz)	*B* (N)
Air-Gap Value		10.164			19.338
Closed-Area Value	0.0406	0.4033	0.0510	0.1139	−1.448

## Data Availability

The code and data used for this research are available publicly at https://github.com/Fworg64/air_gap_coal_sensor_model (Accessed on 28 April 2024, see tag v1.0). https://github.com/Fworg64/Tdataplotter (Accessed on 28 April 2024, see tag v1.0).

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
