# Peer review of "Conforming Capacitive Load Cells for Conical Pick Cutters"

_sensors, 2024, doi:10.3390/s24134238_

Round 1

Reviewer 1 Report

Comments and Suggestions for Authors

The present work attempts to investigate the performance of air-gap capacitive sensing of force on the conical pick cutters. The manuscript contains ample data for the conclusion of the present work. Please find below the following queries:

Comments and Suggestions for Authors:

I. Ideas are not unfamiliar but it is approached in an original way. The results have been appropriately analysed and discussed. It provides a significant new insight in the design and evaluation of force sensing device on conical pick cutters. In Formula (17), σ(·) is not an appropriate expression, since the bariables are usually represented by letters instead of ‘·’. In addition, pressure is often introduced to assess change of the loads applied on capacitive sensors, please explain why force is chosen as the measured parameter.

II. To evaluate the performance of capacitive sensors, The performance variation in capacitive sensors is caused by differences in mechanical and electrical models, processing procedures, and loading conditions. Please tell whether the established measurement model is adaptive and robust, considering different working frequencies (/or working modes) and mechanical characteristics.

III. In the mine environment, whether changes in temperature and humidity can affect the dielectric constant, so as to affect the measured capacitance and measurement accuracy.

        Recommendations:

If it is possible to combine this method with advanced algorithms to establish the relationship between non-electrical parameters and output characteristics, it would be more interesting and bring greater application value.

Comments on the Quality of English Language

The overall English writing of the paper is good, but some grammar and expression need improvement

Author Response

Thank you for your professional review, I have responded to your review in line below:  

>>The present work attempts to investigate the performance of air-gap capacitive sensing of force on the conical pick cutters.

The manuscript contains ample data for the conclusion of the present work. Please find below the following queries:  

>>Comments and Suggestions for Authors:  

>>I. Ideas are not unfamiliar but it is approached in an original way. The results have been appropriately analysed and discussed. It provides a significant new insight in the design and evaluation of force sensing device on conical pick cutters. In Formula (17), σ(·) is not an appropriate expression, since the bariables are usually represented by letters instead of ‘·’.  

Response: Corrections made regarding sigma: the lowercase letter 'z' has been substituted for  ‘·’.  

>>In addition, pressure is often introduced to assess change of the loads applied on capacitive sensors, please explain why force is chosen as the measured parameter.  

Response: Regarding pressure, force, and capacitance: a new paragraph at the end of the results section justifies the selection of force over pressure for rock cutting  

>>II. To evaluate the performance of capacitive sensors, The performance variation in capacitive sensors is caused by differences in mechanical and electrical models, processing procedures, and loading conditions. Please tell whether the established measurement model is adaptive and robust, considering different working frequencies (/or working modes) and mechanical characteristics.  

Response: Additional frequency analysis is presented demonstrating the significant differences in frequency distribution across tool wear and material categories.  Two new paragraphs at the end of the results section (but before the previously mentioned paragraph from point I) describe the differences in frequency distribution.   Additionally, the resonant frequency of the sensor is much greater than the resonant frequency of around 80-100 Hz of the linear cutting machine used in the test. Further filtering of this signal down to 10Hz and the viscous nature of the polyimide greatly reduce the influence of high frequency mechanical ringing. Also, the mentioned frequency based analysis can be used to compensate for bias caused by changing environmental conditions (addressed in the next point).  

>>III. In the mine environment, whether changes in temperature and humidity can affect the dielectric constant, so as to affect the measured capacitance and measurement accuracy.  

Response: Two new paragraphs in the discussion speak to mitigation of changes in performance caused by changing environmental conditions.  

>> Recommendations:

>>If it is possible to combine this method with advanced algorithms to establish the relationship between non-electrical parameters and output characteristics, it would be more interesting and bring greater application value. 

Response: Additional frequency analysis is given to demonstrate the significant differences in response between the conditions of interest, references to prior work on this topic are given.  

Thank you again for your review. Changes in the manuscript are highlighted in green,  except for the following:  

The title has been changed: 

Old: Air-Gap Capacitive Load Cells and Conical Pick Cutters
New: Conforming Capacitive Load Cells for Conical Pick Cutters  

And the abstract has been updated.

Added the following lines as the 6th and 7th sentence of the abstract:   The sensor uses measurements of the resonant frequency of capacitive cells in a steel case to determine cutting forces.  When used in the rock cutting experiments,  the sensor conforms to the tooling  and the stiffness and sensitivity are increased compared to the initial configuration.

Reviewer 2 Report

Comments and Suggestions for Authors

The article has a relevant subject but is written at a low level.

Some points need to be corrected.

Comments:

- Almost all figures in the article are difficult to read and should be redesigned. For example, a Graphical Abstract cannot be understood without the help of the main text of the article.

- It is clear from the article that the task is to improve the performance of an existing sensor by adding an air gap to the sensor. However, the biggest problem with the article is that the authors do not compare numerically the parameters of the old and new sensors, which does not allow us to draw a clear conclusion about the feasibility of this study.

- How does the air gap affect the durability and operation of the sensor?

- What is the impact of the sensor air gap on its cooling and accuracy at different operating temperatures?

Author Response

Thank you for your professional review, I have responded to your comments inline below:  

>>The article has a relevant subject but is written at a low level.

>>Some points need to be corrected.

>>Comments:

>>- Almost all figures in the article are difficult to read and should be redesigned. For example, a Graphical Abstract cannot be understood without the help of the main text of the article.  

Response: The graphical abstract has been updated for clarity; all figures have had font sizes increased for readability.  

>>- It is clear from the article that the task is to improve the performance of an existing sensor by adding an air gap to the sensor. However, the biggest problem with the article is that the authors do not compare numerically the parameters of the old and new sensors, which does not allow us to draw a clear conclusion about the feasibility of this study.  

Response: Clarity is given regarding the intent of this research, the title has been changed to highlight that the load cell in this work conforms to the target tooling, and a numerical comparison between the previous sensors sensitivity and the sensitivity of this one is given at the end of the results section in the last sentence. In the conclusion, it is again mentioned that the sensor conforms to the target tooling when used in the rock cutting application.  

>>- How does the air gap affect the durability and operation of the sensor?  

Response:  The air gap is crushed out during rock cutting, leading to a closed area and a sensor which conforms to the target tooling. After the sensor has conformed,  the air gap is no longer present, mitigating the chances for dust and humidity ingress, and also leaving the sensor in a stiff configuration unlikely to experience further plastic deformation.  In the conclusion, it is also mentioned that this sensor could be used to classify material and tool wear based on the sensed differences in frequency response which are characteristic of the underlying material and tool wear conditions.    

>>- What is the impact of the sensor air gap on its cooling and accuracy at different operating temperatures?  

Response: The air gap is no longer present when used in the rock cutting application, this has been made more clear by changing the title

Old: Air-Gap Capacitive Load Cells and Conical Pick Cutters
New: Conforming Capacitive Load Cells for Conical Pick Cutters  and updating the abstract:

Added the following lines as the 6th and 7th sentance of the abstract:  

The sensor uses measurements of the resonant frequency of capacitive cells in a steel case to determine cutting forces. When used in the rock cutting experiments, the sensor conforms to the tooling  and the stiffness and sensitivity are increased compared to the initial configuration.  

Two new paragraphs at the end of the discussion section address mitigations for changing environmental conditions and how the measurements and methods used can compensate for these types of changes.  

Thank you again for your review. Changes in the manuscript are highlighted in green, except for the above mentioned changes to the title and additions to the abstract.

Round 2

Reviewer 2 Report

Comments and Suggestions for Authors

The authors have improved certain parts of the article. However, the Graphic Abstract remains poorly understood.

Author Response

Thank you for your professional review. Please find below our inline responses to your critiques.

>>The authors have improved certain parts of the article. However, the Graphic Abstract remains poorly understood.

The graphical abstract has been simplified, and the quality of the figure has been improved with photo-quality images.